# Immune Phenotype and Immune Checkpoint Inhibitors for the Treatment of Human Hepatocellular Carcinoma

**DOI:** 10.3390/cancers12051274

**Published:** 2020-05-18

**Authors:** Naoshi Nishida, Masatoshi Kudo

**Affiliations:** Department of Gastroenterology and Hepatology, Kindai University Faculty of Medicine; 377-2 Ohno-Higashi, Osaka-Sayama 589-8511, Japan; m-kudo@med.kindai.ac.jp

**Keywords:** hepatocellular carcinoma, molecular classification, immune phenotype, immune checkpoint inhibitor, stem cell marker, oncogenic signal, β-catenin, genetic alteration

## Abstract

Immunotherapies are promising approaches for treating hepatocellular carcinomas (HCCs) refractory to conventional therapies. However, a recent clinical trial of immune checkpoint inhibitors (ICIs) revealed that anti-tumor responses to ICIs are not satisfactory in HCC cases. Therefore, it is critical to identify molecular markers to predict outcome and develop novel combination therapies that enhance the efficacy of ICIs. Recently, several attempts have been made to classify HCC based on genome, epigenome, and transcriptome analyses. These molecular classifications are characterized by unique clinical and histological features of HCC, as well immune phenotype. For example, HCCs exhibiting gene expression patterns with proliferation signals and stem cell markers are associated with the enrichment of immune infiltrates in tumors, suggesting immune-proficient characteristics for this type of HCC. However, the presence of activating mutations in β-catenin represents a lack of immune infiltrates and refractoriness to ICIs. Although the precise mechanism that links the immunological phenotype with molecular features remains controversial, it is conceivable that alterations of oncogenic cellular signaling in cancer may lead to the expression of immune-regulatory molecules and result in the acquisition of specific immunological microenvironments for each case of HCC. Therefore, these molecular and immune characteristics should be considered for the management of HCC using immunotherapy.

## 1. Introduction

Hepatocellular carcinoma (HCC) remains one of the leading causes of cancer-related morbidity worldwide and generally emerges from a background of chronic liver inflammation [1]. Recent advancements in molecular target therapy have contributed to improvements in the prognosis of HCC patients, even those with advanced disease [2]. However, most cases of HCC show a tolerance or become refractory to molecular target agents during its clinical course [3,4]. On the other hand, immunotherapies are considered to be a promising approach for HCC patients even in those refractory to conventional therapies [5], and several immune components may play a role in the development and progression of this disease [6]. Nevertheless, phase III clinical trials of immune checkpoint monotherapies in patients with HCC have failed to show superiority to control groups for overall survival (OS) and progression-free survival (PFS) [7,8]. 

Several attempts have been made to subclassify HCC based on genetic and epigenetic alterations [9,10,11,12]. It has also been reported that the molecular subclass of HCC sometimes reflects the immune milieu of tumors [13]. For example, an association between molecular alterations of HCC and the expression of immune checkpoint molecules has been reported [14], and alteration of oncogenic signals due to mutations may lead to altered expression of immune modulators [15]. Therefore, a profound understanding of the molecular subclasses that affect the immune status of tumors may provide valuable insight for the rational development of combination therapies using immune checkpoint inhibitors (ICIs). In this review, we focus on this important issue and introduce findings from recent studies regarding molecular classifications and immune phenotype of HCC. Furthermore, we discuss the development of novel combination therapies that may further improve the efficacy of ICIs in these refractory tumors. 

## 2. Molecular Classification and Immune Phenotype of HCC

### 2.1. Oncogenic Signal Activation in HCC

Recent deep sequencing technology has led to the revelation of a complex landscape of genetic alterations in the HCC genome [16,17,18]. Although the majority of the alterations are considered to be passenger mutations that do not affect the immortalization or growth of HCC cells, there are several putative driver mutations that act as gain-of-function or loss-of-function mutations involved in critical signaling pathways [19]. Generally, HCCs develop in livers with chronic damage, such as that caused by hepatitis B virus (HBV), hepatitis C virus (HCV), alcoholic liver disease, and non-alcoholic fatty liver disease (NAFLD). Reportedly, mutations of *CTNNB1* are associated with alcohol intake, while *TP53* mutations are more frequently detected in HBV-positive HCCs than those with other risk factors [20]. However, genes carrying mutations are heterogeneous, regardless of etiology of this type of tumor. Activating mutations in *CTNNB1*, inactivating mutations in *TP53*, and activating mutations in *telomerase reverse transcriptase* (*TERT*) are the most frequently detected mutations in HCC [21,22,23,24,25,26]. Other genetic alterations that lead to constitutive activation of specific growth signals are relatively rare [27]. On the other hand, a considerable percentage of tumors exhibit complex patterns of genetic alterations that lead to the activation and inactivation of various signaling pathways [20]. Guichard et al. classified mutations of HCC based on the signaling pathway involved, such as Wnt/β-catenin, p53/cell cycle control, chromatin remodeling, phosphoinositide 3-kinase (PI3K)/Ras signaling, and oxidative stress and endoplasmic reticulum stress pathway [16]. In addition, constitutive activation of telomerase, which is responsible for the immortalization of cancer cells, may also act as a driver of HCC carcinogenesis [23,24]. 

### 2.2. Molecular Subclass and Tumor Characteristics

Although the genetic alterations and gene expression among individual HCCs are heterogeneous, several studies have classified HCC based on the patterns of these molecular alterations [9,10]. Genetic changes and expression may affect the phenotype of HCC and may be associated with tumor characteristics such as biological behavior [28]. Boyault et al. performed comprehensive analyses of gene expression and classified HCCs using hierarchical clustering analysis. They characterized HCC subclasses according to mutations, chromosomal alterations, copy number of HBV genomes, and DNA methylation of the promoters of *CDH1* and *CDKN2A*. Accordingly, HCCs are subclassified into six groups, with each subclass demonstrating unique molecular characteristics and clinical features [9]. Group 1–3 (G1–G3) tumors are associated with chromosomal instability and amplification and overexpression of cell-cycle/proliferation-related genes, such as *FGF19/CCND1* on 11q13 [9,29,30]. Among these, G1 is characterized by a low copy number of HBV and the expression of genes activated in fetal liver. HCCs of G2 have a high copy number of HBV and mutations in *PIK3CA* and *TP53*. Furthermore, activation of the PI3K-Akt pathway is prominent in both G1 and G2 HCC tumors. Tumors of G3 tend to carry *TSC1/TSC2* mutations. On the other hand, HCCs classified as G4–G6 exhibit low levels of chromosomal alterations. The G4 subtype contains various tumor types with mutations in *TCF1*, while G5 and G6 are strongly correlated to mutations in *CTNNB1*, leading to activation of the Wnt/β-catenin pathway. The *CTNNB1* mutations are frequently accompanied with hypermethylation in the promoter of multiple tumor suppressor genes, especially in HCV-positive and aged patients [22,31]. It has been reported that specific clinical features are associated with different subclasses, such as young age, female, African, and high α-fetoprotein (AFP) with G1, hemochromatosis with G2, and the presence of satellite nodule with G6 [9]. Hoshida et al. also reported an association of molecular features with more aggressive and less-aggressive HCCs, where the aggressive types represented the activation of E2F transcription factor 1 (E2F1) and inactivation of *TP53* [10]. As E2F1 mediates both cell-cycle progression and p53-dependent apoptosis, it is conceivable that the combination of E2F1 activation and p53 inactivation is likely to result in the acceleration of cell cycle progression and tumor growth. These investigators also identified two subclasses of aggressive HCCs (S1 and S2) based on molecular features. The subclass S1 is characterized by activation of the transforming growth factor (TGF)-β pathway and expression of Wnt target genes in the absence of *CTNNB1* mutations. On the other hand, the subclass S2 demonstrates MYC and AKT activation and overexpression of AFP and insulin-like growth factor 2 (IGF2) and is accompanied by the downregulation of interferon (IFN)-related genes. High serum AFP levels, expression of epithelial cell adhesion molecule (EpCAM), and vascular invasion are also frequently observed in S2 HCCs. Expression of stem/biliary markers, such as cytokeratin 19 (CK19), is similarly enriched in both S1 and S2 subclasses. Tumors belonging to subclass S3 are characterized by a less-aggressive phenotype and the retention of mature liver function, as exemplified by the upregulation of genes involved in metabolism, detoxification, and protein synthesis [10]. The activating mutation of *CTNNB1* is primarily observed in S3, which is enriched in the G5 and G6 subclasses of Boyault et al. [9]. 

On the other hand, associations between molecular alteration and clinicopathological characteristics are also reported. Calderaro et al. described the histological features of HCCs that carry *CTNNB1* and *TP53* mutations [32]. *CTNNB1* and *TP53* mutations appear to be mutually exclusive. HCCs with *CTNNB1* mutations are generally large, well-differentiated, and show microtrabecular or pseudoglandular histological patterns, cholestatic tendencies, and a lack of inflammatory infiltrates. On the other hand, *TP53* mutations are associated with poorly differentiated HCCs with a compact pattern, multinucleated and pleomorphic cells, and frequent vascular invasion. These investigators also clarified several molecular characteristics of specific HCC subtypes, including scirrhous subtypes of HCCs that showed *TSC1*/*TSC2* mutations, epithelial-to-mesenchymal transition, and expression of genes related to progenitor cells [32]. The steatohepatitic subtype of HCC is characterized by activation of the interleukin (IL)-6/JAK/STAT pathway with wild-type *CTNNB1*, *TERT*, and *TP53*. Interestingly, such phenotypic features are closely linked to the G1–G6 subgroups proposed by Boyault et al. with the association of progenitor phenotype to G1, macrotrabecular massive subtype and macrovascular invasion to G3, steatohepatitic subtype to G4, and cholestasis and lack of inflammatory infiltrates to G5 and G6 (Figure 1) [9].

Desert et al. further subclassified the non-proliferative phenotype of HCCs that demonstrate a low potential of recurrence [33]. The transcriptomic data revealed two subclasses of non-proliferative HCCs, the periportal-type (wild-type β-catenin) and perivenous-type (mutant β-catenin). HCCs of the periportal-type show activation of a hepatocyte nuclear factor 4A-driven gene, low expression of a metastasis-specific gene, and low frequency of *TP53* mutations. Clinically, such tumors are characterized by early-stage tumors that lack macrovascular invasion. The periportal-type of HCCs represent the gene expression profile, like the S3 signature described by Hoshida et al. Although this type of HCC does not carry mutations in *CTNNB1*, such cases do exhibit a better prognosis than those of the perivenous-type. On the other hand, the perivenous-type tumors have *CTNNB1* mutations that are frequently observed in HCCs, categorized as G5 and G6 (Figure 1) [33]. 

### 2.3. Immune Phenothype of HCC

There are several studies that have clarified the association between immune status and clinical characteristics of HCC, particularly for its relation to the prognosis and the response to the treatment [35,36,37,38]. In addition, recent reports have shown a link between molecular subclass and immune phenotype of HCCs [27,34,39]. Expression of programmed cell death-ligand 1 (PD-L1) in HCC cells is reportedly associated with clinical parameters related to tumor aggressiveness, such as high serum α-fetoprotein levels, satellite nodules, vascular invasion, and poorly differentiated phenotype, as well as molecular features associated with advanced tumor [38,40]. It is also known that PD-L1 expression is more frequently detected in HCCs that express stem/biliary cell markers CK19 and Sal-like protein 4 (SALL4) [34,38,40]. This suggests that PD-L1 expression is associated with the progenitor subtype of HCCs, such as HCCs classified as G1. Furthermore, PD-L1 expression in tumor infiltrates also correlates with aggressive tumor characteristics [38]. 

#### 2.3.1. Classification of HCC Based on the Gene Expression Pattern and Immune Milieu

Sia et al. found that approximately 25% of HCCs they evaluated were classified as “immune-specific class” based on gene expression profiling [39]. Furthermore, they found that this phenotype consists of two immune phenotype subclasses, active and exhausted immune subclasses, according to gene expression profiles of tumor, stromal, and immune cells. The HCCs belonging to the active immune subtype, which is related to better survival, shows enriched gene expression related to antitumor immune response, such as the expression of interferon-related and adaptive immune response genes. In contrast, the exhausted immune subtype enriched with HCCs belonging to the S1 subclass described by Hoshida et al. [10], exhibits gene expression characterized by activation of a potent immunoregulatory cytokine signal, such as transforming growth factor-β (TGF-β), which is known to regulate stroma interactions and angiogenesis, induce T-cell exhaustion, and promote M2 macrophages. Through methylome analyses, it has been suggested that immune subclasses have unique DNA methylation signatures that determine the immune response to HCC [39]. Meanwhile, another study demonstrated that alteration of genes involved in the activation of Wnt/β-catenin signaling results in poorer disease control, shorter PFS, and lower OS with respect to treatment of patients with ICIs [41]. Therefore, the presence of activating mutations involved in Wnt/β-catenin signaling is associated with innate resistant to ICIs. Reportedly, HCCs with a *CTNNB1* mutation show significantly lower enrichment scores for several immune signatures, in particular T cells, and also demonstrate overexpression of *protein tyrosine kinase 2* (*PTK2*), which may lead to immune exclusion [39]. De Galarreta et al. showed that β-catenin-driven tumors are resistant to anti-programed cell death-1 (PD-1) therapy in a mouse model where expression of chemokine (C-C motif) ligand 5 (CCL5) restores immune surveillance [42]. Therefore, an activating mutation in β-catenin may be a negative predictive marker for patients with HCC treated with ICIs.

Immune microenvironment of HCC was classified into three distinct subtypes based on immuohistochemical analyses of the immune regulatory molecules [34]. The subtypes include immune-high, immune-mid, and immune-low groups. HCC classified as the immune-high subtype show increased infiltrations of B cells, plasma cells, and T cells. Consistent with previous reports, the immune-high subtype is characterized by poorly differentiated HCC, positive for CK19 and/or Sal-like protein 4 (SALL4), and enrichment of tumors belonging to S1 and G2 subclasses (Figure 1) [34,38]. It is also confirmed that patients with HCC belonging to the immune-high subtype have better prognosis, even in cases of patients with high-grade tumor. Another study reported that HCCs with immune cell stroma exhibit distinct clinical features of dense CD8^+^ and EBV-positive CD20^+^ tumor infiltrating lymphocytes (TILs) and have good prognosis [43]. This type of HCC is characterized by the lack of *CTNNB1* mutations, global hypermethylation, expression of PD-1 and PD-L1 in tumor infiltrating lymphocytes (TILs), and expression of PD-L1 in tumors. 

Taken together, HCC cases with the immune-high subtype, which is enriched of the tumors with progenitor/proliferative gene expression pattern, especially in the S1 and G2, may also be candidates for treatment with ICIs because this type of HCC generally shows immune infiltrates and express PD-L1 in the tumor tissues. However, a majority of inflammatory infiltrates in tumor show exhausted phenotype with expression of genes involved in Wnt/TGF-β signaling and M2 macrophage [39,40,44,45], and additional agents that alter the immune milieu should be required for the treatment of this subclass. On the other hand, HCCs with expression of adaptive immune response genes, such as IFN-γ, granzyme B, CD8A, and T-cell receptor G, may show a considerable response to ICIs [30,39]. Generally, HCCs with hepatocyte-like/non-proliferative gene expression pattern lack the activation of PD-1/PD-L1 signaling as well as gene expression related to immune infiltrates in the tumor [34,40,41]. 

#### 2.3.2. Characteristics of Inflammatory Infiltrates in HCC Tissues

Expression of immune suppressive receptors in immune infiltrates are associated with shorter survival of patients with HCC. For example, T-cell immunoglobulin and mucin domain 3 (TIM-3) expression in tumor-associated macrophages (TAMs) strongly correlates with higher tumor grade and poor patients’ survival [46], whereas TGF-β induces TIM-3 expression and an alternative activation of macrophages. In addition to TIM-3, expression of another immune suppressive receptor, lymphocyte activation gene-3 (LAG-3), is also increased in immune infiltrates of HCC tissues, suggesting that PD-1, TIM-3, and LAG-3 may cooperate and are implicated in inducing anti-tumor immune tolerance [14,47].

Meanwhile, Zheng et al. characterized the molecular and functional properties of T cells from HCC specimens, adjacent non-tumorous tissues, and peripheral blood using single cell sequencing [44]. In HCCs, T-cell enrichment with clonal expansion of CD8^+^ T-cell populations with exhausted phenotype is observed according to the sequencing of T-cell receptors in TILs. These investigators found that layilin is upregulated in activated CD8^+^ cytotoxic T-cells and regulatory T-cells (Tregs) in HCC and these cells play a role in repression of CD8^+^ T-cell functions. Heterogeneity among the populations of exhausted tumor-infiltrating CD8^+^ T-cells has also been reported. TILs with high expression of PD-1 show higher expression levels of genes that regulate T-cell exhaustion compared to TILs that only moderately express PD-1 [47]. Consistent with another report, cells that express high levels of PD-1 also express TIM-3 and LAG-3 and produce low levels of cytokines necessary for cytotoxic effects of T cells, such as interferon-γ (IFN-γ) and tumor necrosis factor (TNF). In addition, the expression pattern of PD-1 in CD8^+^ TILs characterize the two subgroups of HCCs. HCC tumors with PD-1-high CD8^+^ TILs are more aggressive than those without PD-1-high cells. PD-L1 combined positive score (CPS) can be a biomarker used to predict a favorable response to PD-1/PD-L1 blockade [48]. CPS represents PD-L1 expression in both the tumor and intra-tumor inflammatory cells and is significantly higher in cases with PD-1-high CD8^+^ TILs than those with PD-1-low. Furthermore, incubation of PD-1-high CD8^+^ T-cells from HCCs with anti-PD-1 and anti-TIM-3 or anti-LAG-3 antibodies restore cell proliferation and the production of IFN-γ and TNF-α in response to anti-CD3. Therefore, HCC cases with high expression of PD-1 in CD8^+^ TILs may be good candidates for treatment with a combined immune checkpoint blockade [40,49].

In addition to the exhaustion of CD8^+^ T cells, several stromal cells, such as myeloid-derived suppressor cells (MDSCs), tumor-associated macrophages (TAMs), Tregs, type 2 helper T (Th2) cells, and cancer-associated fibroblasts (CAFs), act in concert in refractoriness to immunotherapy in HCC patients [6,50]. Hypoxia in tumor tissues stimulates the induction of vascular endothelial growth factor (VEGF) in cancer cells and contributes to the recruitment of immune suppressive stromal cells through the binding of VEGF to its receptor on MDSCs, TAMs, Tregs, and CAFs [6,51]. 

So far, infiltration of MDSCs and Tregs is known to be associated with HCC progression and worse outcome of the patients [52,53]. Increase of arginase 1 in MDSC lead to the depletion of arginine, which impairs the function of immune cells [54]. TGF-β and IL-10 from MDSC stimulate Tregs and suppress natural killer (NK) cells [55]. The M2 polarization of macrophages is induced through the secretion of IL-10 from MDSCs, which result in the downregulation of IL-12 in TAMs. High IL-10 and low IL-12 levels further stimulate the induction of Th2 cells and TAM. TGF-β from MDSCs suppress CD4^+^ and CD8^+^ T cells and NK cells. It also induces immune suppressive receptors on T cells and TAMs [46]. TGF-β and IL-10 signaling, along with the stimulation of VEGF signal, play a role for further activation of Tregs [56]. CAFs and endothelial cells can also be players for anti-tumor immunosuppression. Prostaglandin E2 and indoleamine 2,3-dioxygenase (IDO) from CAF lead to the NK cell dysfunction [57]. Endothelial cells in cancer tissues, reportedly, produce the C-X-C motif chemokine ligand 12, resulting in the recruitment of MDSC [58]. Activation of endothelial cells also contribute to the TGF-β-mediated Treg induction.

## 3. Effective Application of Immune Checkpoint Inhibitors for HCC Cases

### 3.1. HCC Response to Immune Checkpoint Inhibitors 

Although several phase II clinical trials of ICI monotherapies have shown favorable outcomes for the use of ICIs in patients with HCC [48,59], a phase III study failed to demonstrate positive results as the first-line treatment with respect to OS and PFS compared to the multi-kinase inhibitor sorafenib [7], and as the second-line treatment after sorafenib compared to best supportive care (Table 1) [8]. However, there are molecular features that may be associated with response to ICIs. For example, the HCC with microsatellite instability is reported to show good response to treatment with pembrolizumab [60]. The presence of *CTNNB1* variants is associated with the activation of Wnt/β-catenin signaling as well as a lack of immune infiltrates in HCC tumors, which are predictors of a poor response to ICIs in patients with HCC [41]. On the other hand, HCC subtypes with high inflammatory infiltrates, such as HCC of the G2 subclass, may be expected for respond to ICIs [34], although additional agents for combination therapy may be required for a good response [40]. Immunohistochemistry-based markers such as CPS may predict the anti-tumor response to ICIs [48]. However, tumor specimens are required in order to perform the immunohistochemical analysis, which are sometimes difficult to obtain in clinical settings. On the other hand, molecular markers based on genetic alterations of tumor cells based on liquid biopsy may be applicable in which DNA from peripheral blood is used for analysis. From this point of view, the development of a mutation-based molecular marker may prove to be a promising approach for identifying responders for ICIs among HCC patients. However, immune infiltrates of tumor tissues frequently express multiple immune checkpoint molecules that are likely to result in refractoriness to immune checkpoint monotherapies [14,34,40]. Therefore, additional agents for combined immune checkpoint blockades should be required to assure improved response rates.

### 3.2. Combined Immune Checkpoint Blockade Based on Inflammatory Infiltrate Characteristics of HCC

As shown above, several studies have analyzed the expression of immune suppressive receptors and ligands in inflammatory infiltrates [14,34,40,45]. Generally, inflammatory cells in HCC express several immunosuppressive molecules, suggesting that such immune cells are functionally compromised. For example, expression of PD-1, TIM-3, LAG-3, and CTLA4 is significantly higher on CD8^+^ and CD4^+^ T-cells in HCC tissue than those in non-tumor tissues or peripheral blood, and dendric cells (DCs), monocytes, and B cells in tumors express ligands for these receptors [45]. In addition, tumor-associated antigen (TAA)-specific CD8^+^ TILs express higher levels of PD-1, TIM-3, and LAG-3 compared to that of other CD8^+^ TILs. Importantly, antibodies against PD-L1, TIM-3, or LAG-3 restore responses of HCC-derived T cells to tumor antigens, and treatment with combinations of these antibodies demonstrate additive effects in the restoration of T-cell function response to TAA [45]. On the other hand, Brown et al. reported the resistance of tumor cells to ICIs through the upregulation of IDO in patients with HCC [61]. Both anti-CTLA4 and anti-PD-1 antibodies induce IDO and the combination of ICIs with 1-methyl-D-tryptophan, an inhibitor of IDO, is able to suppress tumor growth of HCC in a mouse model. Therefore, anti-PD-1 therapy combined with anti-TIM-3, anti-LAG-3, or IDO inhibitor may be worth consideration for patients with HCCs that have exhausted immune infiltrates (Figure 2a). In addition to the phase III combined immune checkpoint blockade using anti-PD-1/PD-L1 and anti-CTLA-4 antibodies, currently, phase I/II clinical trials for the combinations of anti-PD-1 and anti-TIM-3 antibodies (ClinicalTrials.gov NCT03680508), anti-PD-1 and anti-LAG-3 antibodies (NCT03250832), and anti-PD-1 antibody and IDO inhibitors (NCT03695250) are ongoing (Table 2).

### 3.3. Combined Blockade of PD-1/PD-L1 and VEGF Axis

Because HCC is known as a hypervascular tumor where the development of tumor vessels plays an important role in its pathogenesis [62,63], several ongoing clinical studies are evaluating the combination of anti-angiogenic agents and ICIs (Table 3) [64]. Multiple agents that target VEGF and its receptor (VEGFR) are proven to be effective in the treatment of HCC, including the anti-VEGFR2 antibody, ramucirumab [65]. In addition, anti-angiogenic agents are believed to alter the immunosuppressive microenvironment in HCC [6]. It has been reported that anti-angiogenesis normalizes the leaky vascular network induced by VEGF, where the lack of adhesion molecules on endothelial cells may impair the extravasation of T cells [62,66] and induce an immune proficient condition. VEGF play a role in the recruitment of Tregs into tumor tissues and M2 polarization of macrophages via the increase of IL-4 and IL-10. VEGF is also critical for inhibition of the maturation of dendric cells (DCs) by activating NF-κB, production of IDO in tumor cell and macrophage, T-cell exhaustion by inducing PD-1, LAG-3 and TIM-3, accumulation of myeloid-derived suppressor cells (MDSCs), and inhibition of natural killer cell activity [6,67]. Therefore, a combination of ICIs with anti-VEGF agents should be effective (Figure 2b) [67,68,69], although the dosage that best improves the therapeutic effect of ICIs needs to be defined in individual agents [70]. Accordingly, dual blockade of the VEGF/VEGFR and PD-1/PD-L1 axes in patients with advanced HCC using the anti-PD-L1 antibody atezolizumab and the anti-VEGF-A antibody bevacizumab, or the anti-PD-1 antibody camrelizumab and the VRGFR2-TKI apatinib results in considerable ORR (Table 3) [64]. In addition, other combinations modulating immune microenvironment, such as the combination of anti-PD-1 antibody with an inhibitor of TGF-β receptor, is also under the early phase clinical trial (Table 3: NCT02423343).

### 3.4. Immune Checkpoint Inhibitors of Cancer Stem Cells

As previously reported, PD-L1 is expressed in the progenitor subtype of HCCs [34,38]. We also found a significant increase of PD-L1 expression in CK19-positive and/or SALL4-positive HCCs compared to those not expressing such markers [40]. Interestingly, genetic alterations involved in the PI3K-Akt pathway are more frequently detected in PD-L1-positive tumors than in PD-L1-negative tumors [40]. Inactivation of phosphatase and tensin homolog deleted from chromosome 10 (PTEN), which is known to suppress PI3K, leads to the expression of PD-L1 in glioma [71]. More importantly, a recent report suggests that an inactivating mutation of *PTEN* and activating mutation of *PI3KCA* are associated with CK19 expression in HCC [72], where expression of PD-L1 is common. As activation of the PI3K-Akt pathway is a characteristic of cancer stem cells (CSCs) [73], genetic alterations and constitutive activation of this pathway may give rise to the overexpression of PD-L1 and induce stem cell features in HCCs. From this perspective, blockade of the PD-1/PD-L1 axis may be effective for HCC with stem cell-like characteristics, which is resistant to conventional therapies. However, we have also found that infiltration of CD8^+^ cells is not as prominent in PD-L1-positive HCCs with mutations in the PI3K-Akt pathway compared to those without the mutations. Constitutive activation of the PI3K-Akt pathway in HCC might induce PD-L1 expression, even in a non-inflamed background, where a lack of CD8^+^ T-cells could be an obstacle for sufficient action of anti-PD-1/PD-L1 monotherapy. On the other hand, it is also suggested that the PI3K-Akt pathway is frequently activated in CSCs and PI3K inhibitors preferentially target CSCs [73]. As the expression of stem cell markers in HCC is associated with PD-L1 expression and since anti-PD-1/PD-L1 antibody might also target CSCs, a dual blockade of the PD1/PD-L1 axis and PI3K-Akt pathway may be an option for treating patients with HCC showing stem cell features (Figure 2c) [74].

### 3.5. Current Limitation of Immune Checkpoint Inhibitors and Challenge for HCC with Lack of Immune Infiltrates

HCC patients with dense lymphocyte infiltration reportedly show a marked reduction of response rate after curative resection of tumor, suggesting that TILs are critical for anti-tumor immune response [75]. From this point of view, it is conceivable that “immune cold tumor” with lack of immune infiltrates should be refractory to ICIs [66]. Ishizuka et al. reported that loss-of-function of the RNA-editing enzyme adenosine deaminase acting on RNA (ADAR1) overcomes immune checkpoint blockade resistance caused by inactivation of antigen presentation by tumor cells [76]. This restoration of sensitivity to immunotherapy may occur without recognition of TAA by CD8^+^ T-cells. As ADAR1 is able to act as an oncogene and its overexpression plays a role in the carcinogenesis of HCC [77], intervention of ADAR1 activity may also be a promising approach as an effective immunotherapy in patients with HCC refractory to ICIs due to the lack of CD8^+^ TILs (Figure 2d). 

On the other hand, results from methylome analyses of cancer tissues suggest that epigenetic alterations in HCC may affect the anti-tumor immune response. Hong et al. investigated the role of epigenetic therapy on enhancing immunotherapy responses in HCC [78]. Treatment of HCC cell lines with inhibitors of enhancer of zeste homolog 2 (EZH2) and DNA methyltransferase 1 (DNMT1) improved the induction of Th1 chemokines and HCC-related antigens upon treatment with anti-PD-L1 antibody. Furthermore, using an in vivo model, they found that the combination of PD-1/PD-L1 blockade with an epigenetic modulator improves the trafficking of CD8^+^ T-cells into tumor tissues and promotes tumor regression. Therefore, epigenetic modulation may reactivate the epigenetically repressed chemokine responsible for T-cell trafficking and induce neoantigens as immune targets. Thus, the combination of epigenetic therapy with ICIs might also be applicable to cases with refractory HCC (Figure 2d). Schonfeld et al. showed that polymorphism in the protein arginine methyltransferase 1 (PRMT1) was associated with protein expression and modulated the expression of PD-L1 and PL-L2 in HCC cells [79], suggesting that intervention of PRMT1 activity could also restore the response to immune checkpoint inhibitors in some patients.

For the development of biomarkers that predict the tumor response to immunotherapy, it is critical to improve the outcome of the treatment. Previous reports point out that tumors with active IFN-γ signaling show immune classes that can be candidates for immunotherapy [30,39]. In addition, expression of PD-L1 in tumor cells and tumor infiltrates (CPS) was reportedly associated with tumor response in HCC cases [48]. Detection of activating mutation in *CTNNB1* should also be informative to know immune cold phenotype and lack of response to ICIs in HCC [41]. On the other hand, Feun et al. indicated that baseline plasma TGF-β level could be a predictive biomarker for the response to pembrolizumab [80], and clinical trials of combined blockade of PD-1/PD-L1 and TGF-β axis are ongoing (Table 3). Dong et al. analyzed multiple tumors of the same patients for genetic structure, neoantigens, T cell receptor repertoires, and immune infiltrates, and found that only a few tumors were under the control of immunosurveillance and the majority carry a variety of immune escape mechanisms, even in a single case [81]. From this point of view, precise analysis of immune phenotype of HCC should contribute to the establishment of personalized immunotherapy in HCC cases.

## 4. Conclusions

Several studies have demonstrated the efficacy of immune checkpoint inhibitors in HCC, even in tumors that are resistant to conventional therapies. However, only a small subset of HCCs show an anti-tumor response to immune checkpoint monotherapy [5,48,59,82]. Therefore, understanding the immunological microenvironment of HCC is crucial since the response to anti-PD-1 therapy may be determined by the immune status of the tumor [13,15,83]. As the mutational signature of HCC may affect its immunophenotype thorough the induction of immune regulatory molecules and cells, the data presented here may be informative in the development of effective combination therapies using ICIs for treating patients with HCC, especially those who are refractory to conventional therapies.

## Figures and Tables

**Figure 1 cancers-12-01274-f001:**
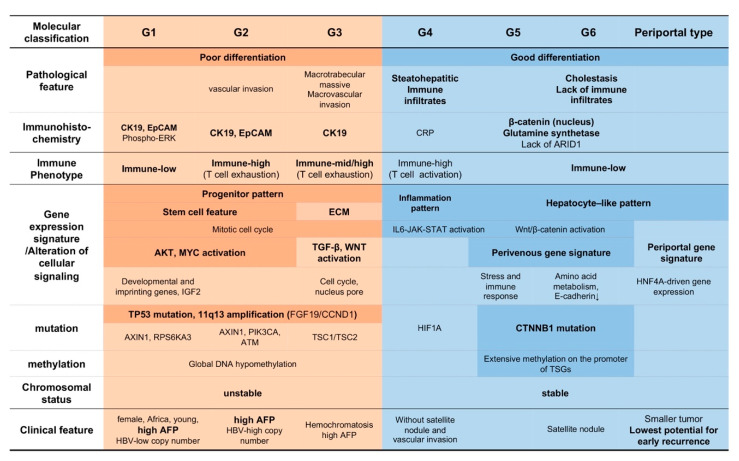
Molecular classification, clinicopathological characteristics, and immune phenotype of human hepatocellular carcinoma (HCC). Subclasses shown as G1–G6 were described by Boyault et al. [9]. The associations between the molecular subclass and pathological characteristics were reported by Calderaro et al. [32]. The classification shown as “periportal type” was described by Desert et al. [33]. Immune phenotype in this figure (immune-high, -med, and -low) was proposed by Kurebayashi et al. [34]. The bold denotes the representative findings of molecular, clinical, and pathological features. ECM: extracellular matrix. HNF4A: hepatocyte nuclear factor 4A. HIF1A: hypoxia inducible factor A. TSGs: tumor suppressor genes.

**Figure 2 cancers-12-01274-f002:**
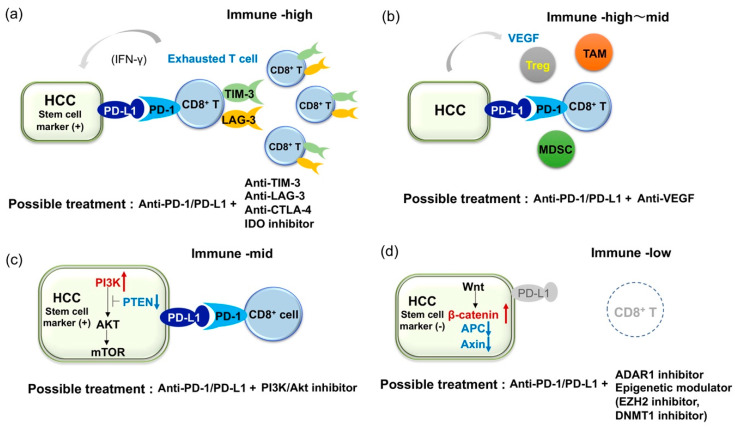
Illustrative figures of expected combination therapies for HCC patient refractory to immune checkpoint monotherapies. (**a**) In cases with expression of PD-L1 in HCC but multiple co-inhibitory receptors on tumor infiltrates, dual blockade of PD-1/PD-L1 and anti-TIM-3 or anti-LAG-3 should be required. (**b**) Because VEGF is known to play an important role for induction of immune suppressive molecules and cells, dual blockade of PD-1/PD-L1 and VEGF axis should be effective. (**c**) In cases with expression of PD-L1 and activating mutation in the PI3K-mTOR pathway in HCC, dual blockade of PD-1/PD-L1 and the PI3K-mTOR pathway might be effective. Notably, both anti-PD-1/PD-L1 and anti-PI3K-mTOR agents could target cancer stem cells (CSCs). (**d**) In cases with a lack of CD8^+^ T cell infiltration in tumor (activating mutation in the β-catenin pathway is common in this type), ADAR1 inhibitor and epigenetic modulator might induce the recruitment of CD8^+^ T cells into tumor and contribute to the induction of anti-tumor immunity.

**Table 1 cancers-12-01274-t001:** Clinical trials and outcomes of immune checkpoint monotherapies in HCC.

Clinical Trial ID	Trial Name	Agents ^1^	Setting ^2^	Key Outcome ^3^
***Phase I/II***				
NCT01658878	CheckMate 040	**Nivolumab**	dose-escalation, n = 48, dose-expansion, n = 214	**ORR: 20% ^4^** DCR: 64%, (37%) ^5^ OS: 13.2 months (8.6–NE) ^6^
NCT02702414	KEYNOTE-224	**Pembrolizumab**	second-line n = 104	**ORR: 17% ^7^** DCR: 62% OS: 12.9 months (9.7–15.5)
***Phase III***				
NCT03383458	CheckMate 9DX	**Nivolumab** versus placebo	adjuvant, randomized, double-blinded (n = 530)	**RFS**
NCT02576509	CheckMate 459	**Nivolumab** versus Sorafenib	first-line, randomized, open label, n = 743	**Median OS: 16.4 months in the****nivolumab****group and 14.7 months in the sorafenib group. ^8^** Median PFS: 3.7 months for nivolumab and 3.8 months for sorafenib. ORR: 15% in the nivolumab group and 7% in the sorafenib group.
NCT03412773	Rationale-301	**Tislelizumab** versus sorafenib	first-line, randomized, open label, (n = 674)	**OS**
NCT02702401	KEYNOTE-240	**Pembrolizumab** versus placebo	second-line, randomized, double-blinded, n = 413	**Median OS: 13.9 months in the pembrolizumab group and 10.6 months in the placebo group; HR 0.781, *p* = 0.0238.** **Median PFS: 3.0 months for pembrolizumab and 2.8 months for placebo; HR 0.781, *p* = 0.0022. ^9^** ORR: 18.3%, DCR: 62.2%

^1^ Bold denotes immune checkpoint inhibitors. ^2^ n, number of the patients analyzed in the study. The number in the parenthesis shows the number of the planned enrollment. ^3^ Bold denotes the primary outcome measures of the study. Duration of responses and survival are shown as median values. The numbers in the parenthesis show 95% confidential interval (CI). ^4^ El-Khoueiy et al. Lancet 2017; 389: 2492–2502 [59]. ^5^ Disease control with stable disease for ≥6 months. ^6^ Median overall survival of the sorafenib progressor without viral hepatitis in the dose-expansion cohort. ^7^ Zhu et al. Lancet Oncol 2018; 19: 940–952 [48]. ^8^ Yau et al. The European Society for Medical Oncology (ESMO) 2019 congress (# LBA38). ^9^ Finn et al. J Clin Oncol 2019; 38: 193–202 [8]. The 95% CI of median OS: 11.6 to 16.0 months in the pembrolizumab group and 8.3 to 13.5 months in the placebo group (hazard ratio, HR, 0.781; 95% CI, 0.611 to 0.998; *p* = 0.0238). The 95% CI of median PFS was 2.8 to 4.1 months for pembrolizumab and 1.6 to 3.0 months for placebo (HR, 0.718; 95% CI, 0.570 to 0.904; *p* = 0.0022). OS and PFS did not reach statistical significance per specified criteria in this study. ORR, objective response rate; DCR, disease control rate; OS, overall survival; NE, not estimated; RSF, recurrence-free survival; PFS, progression-free survival.

**Table 2 cancers-12-01274-t002:** Clinical trials and outcomes of combined immune checkpoint blockade in HCC.

Clinical Trial ID	Trial Name	Agents ^1^	Setting ^2^	Key Outcome ^3^
***Phase I/II***				
NCT01658878	CheckMate 040	**Nivolumab + Ipilimumab**	n = 50	**ORR: 32% ^4^** DCR: 54% OS: 22.8 months (9.4–NE) DOR: 17.5 months (4.6–30.5)
NCT02519348		**Durvalumab ± Tremelimumab**	n = 40	ORR: 25% ^5^ DCR: 57.5%
NCT03680508		**TSR-002 + TSR-042 (Dostarlimab)**	first-line, (n = 42)	**ORR**
NCT03250832		**TSR-033 + TSR-042**	dose escalation and dose expansion cohorts (n = 200)	**AEs for** dose escalation cohort **ORR for** dose expansion cohort
NCT03695250		**BMS986205 + Nivolumab**	first- or second-line, (n = 23)	**AEs and ORR**
***Phase III***				
NCT04039607	CheckMate9DW	**Nivolumab + Ipilimumab** versus Sorafenib/Lenvatinib	first-line, randomized, open label, (n = 1084)	**OS**
NCT03298451	HIMARAYA	**Durvalumab ± Tremelimumab** versus Sorafenib	first-line, randomized, open label, (n = 1310)	**OS**

^1^ Bold denotes immune checkpoint inhibitors. ^2^ n, number of the patients analyzed in the study. The number in the parenthesis shows the number of the planned enrollment. ^3^ Bold denotes the primary outcome measures of the study. Duration of responses and survival are shown as median values. The numbers in the parenthesis show 95% confidential interval. ^4^ Yau et al. J Clin Oncol. 2019; 37 (supplement abstract 4012). ^5^ Kelley et al. J Clin Oncol 2017; 35 (supplement abstract 4073). DOR, duration of response; AEs, adverse events.

**Table 3 cancers-12-01274-t003:** Clinical trials and outcomes of the combination therapies with immune checkpoint inhibitors and molecular targeted agents.

Clinical Trial ID	Trial Name	Agents ^1^	Setting ^2^	Key Outcome ^3^
***Phase I/II***				
NCT03299946	CaboNivo	Cabozantinib + **Nivolumab**	neoadjuvant, (n = 15)	**AEs and number of patients who complete the treatment.**
NCT03006926		Lenvatinib + **Pembrolizumab**	first-line, (dose-escalation, dose-expansion), n = 30 (n = 97)	**ORR: 53.3% (34.3–71.7), DOR: 8.3 months (3.8–11.0) ^4^** DCR = 90.0%; 73.5–97.9, PFS: 9.7 months 7.7–NE, OS: 14.6 months 9.9–NE.
NCT03289533	VEGF Liver 100	**Avelumab** + Axitinib	AFP ≥400 ng/mL, n = 22	**AE** ORR: 13.6% (2.9–34.9) ^5^ DCR: 68.2 (45.1–86.1) PFS: 5.5 months (1.9–7.4) OS: 12.7 months (0.0–NE) DOR: 5.5 months (3.7–7.3)
NCT03418922		Lenvatinib + **Nivolumab**	first-line, (n = 30)	**DLT, AEs**
NCT02715531	GO30140	**Atezolizumab** + Bevacizumab	n = 73	**ORR: 27% ^6^** **PFS: 7.5 months (0.4–23.9+)**
NCT01658878	CheckMate 040	Cabozantinib + **Nivolumab ± Ipilimumab**	first or second-line, (dose-escalation, dose-expansion), (n = 1097, across all cohorts)	**safety, tolerability, ORR**
NCT03170960	COSMIC-021	Cabozantinib + **Atezolizumab**	first-line, (dose-escalation and dose-expansion), (n = 1732, across all cohorts)	**MTD, ORR**
NCT03347292		Regorafenib + **Pembrolizumab**	first-line, (dose-escalation and dose-expansion, n = 57)	**TEAE, DLT**
NCT03539822	CAMILLA	Cabozantinib + **Durvalumab**	second-line, (n = 30)	**MTD**
NCT03475953	REGOMUNE	Regorafenib + **Avelumab**	Second-line, (n = 212)	**Recommended phase II dose, ORR**
NCT02572687		Ramucirumab + **Durvalumab**	Second-line and AFP ≥1.5x ULN, n = 28	**DLTs** ORR: 11% ^7^ PFS: 4.4 months (1.6–5.7) OS: 10.8 months (5.1–18.4)
NCT3463876	RESCUE	**SHR-121 (****Camrelizumab)** + Apatinib	n = 18 (n = 40)	**ORR: 38.9%**^8^ DCR: 83.3% PFS: 7.2 months (2.6–NE)
NCT02423343		Galunisertib (TGFβ receptor I inhibitor) + **Nivolumab**	second-line and AFP ≥200 ng/mL, (dose escalation and cohort expansion, n = 75)	**MTD**
***Phase III***				
NCT03847428	EMERALD-2	**Durvalumab** ± Bevacizumab versus placebo	adjuvant, randomized, double-blinded, (n = 888)	**RFS**
NCT03434379	IMbrave150	**Atezolizumab** + Bevacizumab versus sorafenib	first-line, randomized, open label, n = 501	**OS:****not reached for****Atezolizumab + bevacizumab****vs 13.2 months for sorafenib; HR 0.58, *p* = 0.006 ^9^** **PFS: 6.8 months****for****Atezolizumab + bevacizumab versus 4.3 months for sorafenib; HR 0.59, *p* < 0.0001** ORR: 27%
NCT03713593	LEAP-002	Lenvatinib + **Pembrolizumab** versus Lenvatinib	first-line, randomized, double-blinded, (n = 750)	**OS, PFS**
NCT03755791	COSMIC-312	Cabozantinib + **Atezolizumab** versus Sorafenib versus Cabozantinib	first-line, randomized, open label, (n = 740)	**OS, PFS**

^1^ Bold denotes immune checkpoint inhibitors. ^2^ n, number of the patients analyzed in the study. The number in the parenthesis shows the number of the planned enrollment. ^3^ Bold denotes the primary outcome measures of the study. Duration of responses and survival are shown as median values. The numbers in the parenthesis show 95% confidential interval. ^4^ Ikeda et al. The American Association for Cancer Research (AACR) annual meeting 2019 (abstract #18). ^5^ Mudo et al. J. Clin Oncol 2019; 37 (supplement. abstract 4072). ^6^ Pishvaian et al. ESMO 2018 congress (# LBA26). ^7^ Bang et al. J Clin Oncol 2019; 37 (supplement. abstract). ^8^ Xu et al. J Clin Oncol 2018; 36 (supplement. abstract 4075). ^9^ Cheng et al. ESMO Asia2019 congress (# LBA3). DLT, dose-limiting toxicity; MTD, maximum tolerated dose; TEAEs, treatment-emergent adverse event.

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
