# Peer review of "Immune Phenotype and Immune Checkpoint Inhibitors for the Treatment of Human Hepatocellular Carcinoma"

_cancers, 2020, doi:10.3390/cancers12051274_

Round 1
Reviewer 1 Report
Review article by Nishida and Kudo discusses the Molecular characteristics and immune phenotype of Hepatocellular carcinoma. Manuscript was well written. Some minor editorial corrections are required.
The title of the review article seems to be too broad for its current contents. If authors plan to select this title, the information provided in this review seems to be very limited. If there is no restriction on word limit, authors should elaborate the discussions on each sections with more recent references. If authors could discuss molecular characteristics and immune phenotype separately would be more interesting to read. Otherwise it is confusing and flow is limited. In the current version, even though there are subtitles, content of the sections seems to include both molecular characteristics and immune phenotype (eg. 2.3.1). Also, authors should briefly discuss different cell types and not just T cells (current understanding of T-cells, NK cells, tumor associated macrophages and neutrophils in defining the immune phenotype and immune therapeutic targeting of HCC).
Section 3.1: It is good to summarize current status of immunotherapy clinical trials and their known outcome in a table form.
Section 3.2 on immune check point looks good.
If authors could include a section on challenges and limitations of immuno-therapy especially in combination would have been good.
Some additional references that may be discussed
J Hepatol. 2020 Feb;72(2):215-229. doi: 10.1016/j.jhep.2019.08.017.Advances in molecular classification and precision oncology in hepatocellular carcinoma.
J Hepatol. 2020 Feb;72(2):342-352. doi: 10.1016/j.jhep.2019.09.010.Molecular therapies for HCC: Looking outside the box.
Nat Genet 2015;47:505‐511. chulze K, Imbeaud S, Letouze E, Alexandrov LB, Calderaro J, Rebouissou S, et al. Exome sequencing of hepatocellular carcinomas identifies new mutational signatures and potential therapeutic targets.
NatGenet, 18 (1998),pp. 65-68, 10.1038/ng0198-65 J. Nakayama, H. Tahara, E. Tahara, M. Saito, K. Ito, H. Nakamura, et al.Telomerase activation by hTRT in human normal fibroblasts and hepatocellular carcinomas
NatCommun, 4 (2013),p. 2218, 10.1038/ncomms3218J.C. Nault, M. Mallet, C. Pilati, J. Calderaro, P. Bioulac-Sage, C. Laurent, et al.High frequency of telomerase reverse-transcriptase promoter somatic mutations in hepatocellular carcinoma and preneoplastic lesions,
J Biol Chem. 2020 Apr 3. pii: jbc.RA120.013401. doi: 10.1074/jbc.RA120.013401. [Epub ahead of print]. The polymorphism rs975484 in the protein arginine methyltransferase-1 gene modulates expression of immune checkpoint genes in hepatocellular carcinoma
J Hepatol. 2019 Dec 27. pii: S0168-8278(19)30759-7. doi: 10.1016/j.jhep.2019.12.014. [Epub ahead of print]. Heterogeneous immunogenomic features and distinct escape mechanisms in multifocal hepatocellular carcinoma.
Gastroenterology. 2017 Oct;153(4):1107-1119.e10. doi: 10.1053/j.gastro.2017.06.017. Epub 2017 Jun 23. Antibodies Against Immune Checkpoint Molecules Restore Functions of Tumor-Infiltrating T Cells in Hepatocellular Carcinomas.
Author Response
Response to the Reviewer 1 comments:
Point 1
Review article by Nishida and Kudo discusses the Molecular characteristics and immune phenotype of Hepatocellular carcinoma. Manuscript was well written. Some minor editorial corrections are required.
The title of the review article seems to be too broad for its current contents. If authors plan to select this title, the information provided in this review seems to be very limited. If there is no restriction on word limit, authors should elaborate the discussions on each sections with more recent references. If authors could discuss molecular characteristics and immune phenotype separately would be more interesting to read. Otherwise it is confusing and flow is limited. In the current version, even though there are subtitles, content of the sections seems to include both molecular characteristics and immune phenotype (eg. 2.3.1).
Response 1:
Thank you for the valuable suggestion regarding the construction of the review. We learned the comment carefully and corrected the manuscript. We mainly focused on the immunophenotype and immune checkpoint inhibitors (recent advancement, current limitation, and challenge) for HCC treatment. For this purpose, we changed the title as "Immunophenotype and immune checkpoint inhibitors for the treatment of human hepatocellular carcinoma", which should better reflect the contents. We also corrected the subtitle of 2.3.1 as "Classification of HCC based on the gene expression pattern and immune milieu", and included the additional statement regarding the specific gene expressions and immune subtype in page 5, line 203 - page 6, line 210.
Point 2
Also, authors should briefly discuss different cell types and not just T cells (current understanding of T-cells, NK cells, tumor associated macrophages and neutrophils in defining the immune phenotype and immune therapeutic targeting of HCC).
Response 2:
We agree the discussion regarding the role of cellular component on immunological microenvironment is important. We showed the discussion of cellular components, MDSCs, Tregs, Th2 cells, TAMs, CAFs, and NK cells, in defining the immune phenotype in page 6, line 242 - page 7, line 261.
Point 3
Section 3.1: It is good to summarize current status of immunotherapy clinical trials and their known outcome in a table form.
Response 3:
We summarize current status of immunotherapy clinical trials and outcomes in 3 tables.
Table 1: Clinical trials and outcomes of immune checkpoint monotherapies in HCC. (Page 7, line 285 - page 9, line 303).
Table 2: Clinical trials and outcomes of combined immune checkpoint blockade in HCC. (Page 9, line 329 - 338).
Table 3: Clinical trials and outcomes of the combination therapies with immune checkpoint inhibitors and molecular targeted agents. (Page 10, line 361 - page 11, line 375).
We added the additional explanation for table 2 in page 8, line 321-325, and for table 3 in page 10, line 358-360.
Point 4
Section 3.2 on immune check point looks good. If authors could include a section on challenges and limitations of immuno-therapy especially in combination would have been good.
Response 4:
Mainly, we discussed the challenges and limitations of immuno-therapy in session 3.5 "Current limitation of immune checkpoint inhibitors and challenge for HCC with lack of immune infiltrates". For this purpose, we changed the subtitles that best represent the discussion. Additional discussion was presented in page 12, line 400-402, page 12, line 422 - page 13, line 440, and page 9, line 353-354.
Point 5
Some additional references that may be discussed
J Hepatol. 2020 Feb;72(2):215-229. doi: 10.1016/j.jhep.2019.08.017.Advances in molecular classification and precision oncology in hepatocellular carcinoma.
J Hepatol. 2020 Feb;72(2):342-352. doi: 10.1016/j.jhep.2019.09.010.Molecular therapies for HCC: Looking outside the box.
Nat Genet 2015;47:505‐511. chulze K, Imbeaud S, Letouze E, Alexandrov LB, Calderaro J, Rebouissou S, et al. Exome sequencing of hepatocellular carcinomas identifies new mutational signatures and potential therapeutic targets.
NatGenet, 18 (1998),pp. 65-68, 10.1038/ng0198-65 J. Nakayama, H. Tahara, E. Tahara, M. Saito, K. Ito, H. Nakamura, et al.Telomerase activation by hTRT in human normal fibroblasts and hepatocellular carcinomas
NatCommun, 4 (2013),p. 2218, 10.1038/ncomms3218J.C. Nault, M. Mallet, C. Pilati, J. Calderaro, P. Bioulac-Sage, C. Laurent, et al.High frequency of telomerase reverse-transcriptase promoter somatic mutations in hepatocellular carcinoma and preneoplastic lesions,
J Biol Chem. 2020 Apr 3. pii: jbc.RA120.013401. doi: 10.1074/jbc.RA120.013401. [Epub ahead of print]. The polymorphism rs975484 in the protein arginine methyltransferase-1 gene modulates expression of immune checkpoint genes in hepatocellular carcinoma
J Hepatol. 2019 Dec 27. pii: S0168-8278(19)30759-7. doi: 10.1016/j.jhep.2019.12.014. [Epub ahead of print]. Heterogeneous immunogenomic features and distinct escape mechanisms in multifocal hepatocellular carcinoma.
Gastroenterology. 2017 Oct;153(4):1107-1119.e10. doi: 10.1053/j.gastro.2017.06.017. Epub 2017 Jun 23. Antibodies Against Immune Checkpoint Molecules Restore Functions of Tumor-Infiltrating T Cells in Hepatocellular Carcinomas.
Response 5:
Thank you for the suggestions for important references. We added the discussion for all of these 8 papers in the text. In addition, we included additional 12 references for the revision.
We also make minor alterations in figure 1 and figure 2.

Reviewer 2 Report
Immune checkpoint inhibitors (ICIs) provide a promising approach for treatment of hepatocellular carcinoma (HCC). Although this therapeutic approach is promising, the response rate for these ICIs remains unsatisfactory. Therefore, it is crucial to identify the molecular markers to predict outcome for personalized medicine and ultimately develop combination strategy to enhance the treatment efficacy. In this review article, Nishida et al. first provided molecular classification, clinicopathological characteristics and immune phenotype of HCC. They further characterized the types of inflammatory infiltrates in HCC. In addition, they provided some strategy to overcome of Resistance to immunotherapy in HCC with the Lack of infiltrating immune cells. They have also provided a comprehensive discussion on the key oncogenic signaling pathways and stem cell markers which are enriched with immune infiltrates. Lastly, they discussed the potential combination strategies to enhance the efficacy of current ICIs in HCC treatment. This review is very comprehensive, up-to-dated and provides new direction for the current ICI treatment in HCC. In order to enhance the content of this article, I would suggest to provide a summary table showing the data of current clinical trials in different clinical centers so that the readers will understand better for the current efficacy of ICI (PD1) in HCC such as response rate, etc.
Author Response
Response to the Reviewer 2 comments
Point 1
This review is very comprehensive, up-to-dated and provides new direction for the current ICI treatment in HCC. In order to enhance the content of this article, I would suggest to provide a summary table showing the data of current clinical trials in different clinical centers so that the readers will understand better for the current efficacy of ICI (PD1) in HCC such as response rate, etc.
Response 1:
Thank you for the important and positive comments.
We summarize current status of immunotherapy clinical trials and outcomes in 3 tables.
Table 1: Clinical trials and outcomes of immune checkpoint monotherapies in HCC. (Page 7, line 285 - page 9, line 303).
Table 2: Clinical trials and outcomes of combined immune checkpoint blockade in HCC. (Page 9, line 329 - 338).
Table 3: Clinical trials and outcomes of the combination therapies with immune checkpoint inhibitors and molecular targeted agents. (Page 10, line 361 - page 11, line 375).
We added the additional explanation for table 2 in page 8, line 321-325, and for table 3 in page 10, line 358-360.

Reviewer 3 Report
This is an informative review on what is available in literatures on immune phenotypes in hepatocellular carcinoma. The authors are clearly experts in the field. This review is useful to give insights on the current trends, obstacles, and molecular background (e.g. HCC vast heterogeneity and gene signature) which might hamper the success of immunotherapy against HCC.
Author Response
Response to the Reviewer 3 comment
Point 1
This is an informative review on what is available in literatures on immune phenotypes in hepatocellular carcinoma. The authors are clearly experts in the field. This review is useful to give insights on the current trends, obstacles, and molecular background (e.g. HCC vast heterogeneity and gene signature) which might hamper the success of immunotherapy against HCC.
Response 1:
Thank you for the valuable comments for the manuscript. We hope and believe this review is informative for many readers.

Round 2
Reviewer 1 Report
Authors have addressed my comments which has improved the manuscript.